# Site-selective chemical reactions by on-water surface sequential assembly

Anupam Prasoon [1,2,10], Xiaoqing Yu [3,10], Mike Hambsch[4], David Bodesheim [5], Kejun Liu[1], Angelica Zacarias [2], Nguyen Ngan Nguyen[1], Takakazu Seki [3], Aerzoo Dianat [5], Alexander Croy [6], Gianaurelio Cuniberti [5,7], Philippe Fontaine [8], Yuki Nagata [3], Stefan C. B. Mannsfeld [4] ✉, Renhao Dong [1,9] ✉, Mischa Bonn [3] ✉ & Xinliang Feng [1,2] ✉

Controlling site-selectivity and reactivity in chemical reactions continues to be a key challenge in modern synthetic chemistry. Here, we demonstrate the discovery of site-selective chemical reactions on the water surface via a sequential assembly approach. A negatively charged surfactant monolayer on the water surface guides the electrostatically driven, epitaxial, and aligned assembly of reagent amino-substituted porphyrin molecules, resulting in a well-defined J-aggregated structure. This constrained geometry of the porphyrin molecules prompts the subsequent directional alignment of the perylenetetracarboxylic dianhydride reagent, enabling the selective formation of a one-sided imide bond between porphyrin and reagent. Surface-specific in-situ spectroscopies reveal the underlying mechanism of the dynamic interface that promotes multilayer growth of the site-selective imide product. The site-selective reaction on the water surface is further demonstrated by three reversible and irreversible chemical reactions, such as imide-, imine-, and 1, 3-diazole (imidazole)- bonds involving porphyrin molecules. This unique sequential assembly approach enables site-selective chemical reactions that can bring on-water surface synthesis to the forefront of modern organic chemistry.

In modern synthetic chemistry, controlling selectivity is crucial yet extremely challenging. Selectivity requires a reactant molecule to be inserted at a specifically desired position to favor one product over the other[1,2]. Chemical selectivity can be loosely classified into three categories:[1,3] (i) chemo-selectivity arising from chemical reactivity, viz. the preferential reactivity of one specific functional group over another[4,5], (ii) diastereo-selectivity and enantio-selectivity arising from the spatial arrangement which leads to the predominant formation of

[1]Center for Advancing Electronics Dresden (cfaed) and Faculty of Chemistry and Food Chemistry, Technische Universität Dresden, 01062 Dresden, Germany. [2]Max Planck Institute for Microstructure Physics, Halle (Saale) D-06120, Germany. [3]Max Planck Institute for Polymer Research, Ackermannweg 10, 55128 Mainz, Germany. [4]Center for Advancing Electronics Dresden (cfaed) and Faculty of Electrical and Computer Engineering, Technische Universität Dresden, 01062 Dresden, Germany. [5]Institute for Materials Science and Max Bergmann Center of Biomaterials, Technische Universität Dresden, 01062 Dresden, Germany. [6]Institute of Physical Chemistry, Friedrich Schiller University Jena, 07737 Jena, Germany. [7]Dresden Center for Computational Materials Science (DCMS), Technische Universität Dresden, 01062 Dresden, Germany. [8]Synchrotron SOLEIL, L'Orme des Merisiers, Département 128, 91190 Saint-Aubin, France. [9]Key Laboratory of Colloid and Interface Chemistry of the Ministry of Education, School of Chemistry and Chemical Engineering, Shandong University, Jinan 250100, China. [10]These authors contributed equally: Anupam Prasoon, Xiaoqing Yu. ✉e-mail: stefan.mannsfeld@tu-dresden.de; renhaodong@sdu.edu.cn; bonn@mpip-mainz.mpg.de; xinliang.feng@tu-dresden.de

one diastereomer/enantiomer over the other[6,7], and (iii) regio-selectivity or site-selectivity arising from the specific orientation of the molecules and its chemical reactivity[8,9]. The terms regio- and site-selectivity are often interchangeably used in modern synthetic chemistry, where one direction/site of bond formation/cleavage occurs preferentially over other possible directions/sites[10–12]. The complex organic molecules that play a vital role in living organisms and are also commonly used in medicinal chemistry typically contain multiple functional groups with similar chemical reactivities under the same chemical environments and, therefore, tend to undergo the same chemical transformation pathway under given reaction conditions[13–15]. Thus, identifying and/or designing the optimal combination of chemical reactants, catalysts, and reaction conditions along with a broad range of chemical reactions and diverse substrates that enables the selective targeting of a single site in a large number of similar available sites in a given molecule, with extremely high site-selectivity, represents an enormous synthetic challenge.

Control over the chemo-, diastereo- and enantio-selectivity has been extensively explored in the literature[16–19]. However, the advancement of site-selective reactions (SSRs) appears to be lagging[2,11,20]. Various approaches have been implemented to confer site-selectivity, which include (i) innovative design of selective catalysts, used to induce reaction at the specific sites[9,21–23], (ii) establishment of alternative reaction pathways through weak bonding strategy[24,25], (iii) strategies using protective/deprotective groups to shield undesired reactive sites[26–28], (iv) introduction of steric hindrance/bulkiness within the molecular structure[29,30], and many more as elegantly summarized in recent reviews[8,31–34]. Very recently, the site-selective reaction on a metal surface was achieved by introducing a dissymmetric binding strategy[35], and a stepwise activation strategy to the on-surface chemistry under ultrahigh vacuum (UHV) conditions[36] where substrate-reagent pre-assembly with lattice-matching/mismatching played an important role in lowering the entropic barriers of the reactivity for a given specific site. Good control over the site-selectivity and reactivity demands a deeper mechanistic understanding of site-selective chemical reactions, which opens up a new avenue in synthetic, biological, and materials chemistry[8,37–39]. Thus, new methodologies to achieve unique reactivity and selectivity in a chemical reaction are highly desirable.

Herein, we present the discovery of site-selective chemical reactions with a wide range of chemical reaction types, including imide-, imine-, and 1,3-diazole (imidazole)- bond formations, via a sequential assembly approach assisted by a charged surfactant monolayer on the water surface. By engineering the interfacial environment in on-water chemical reactions, reactant molecules can be made to behave very differently at the water surface than in the bulk phase. We observed the pre-organization of reagent amino-substituted porphyrin molecules into a unique J-aggregated packing structure on the water surface underneath the surfactant monolayer in an epitaxial manner[40–42]. This constrained structure facilitates the precise positioning of the perylenetetracarboxylic dianhydride reagent in a particular direction, leading to the completely selective formation of a one-sided imide bond instead of a two-sided imide product. The step-wise mechanism and multilayer growth of imide site-selective product on the water surface was further studied by using different in-situ techniques such as sum frequency generation (SFG) spectroscopy and grazing incidence X-ray diffraction (GIXD), along with theoretical calculations.

## Results
### Site-selective sequential chemical reactions on the water surface
Site-selective compounds, SSC-1 ((9-(4-(10,15,20-triphenylporphyrin-5-yl)phenyl)−1H isochromeno [6′,5′,4′:10,5,6]anthra[2,1,9-def]isoquinoline-1,3,8,10(9H)-tetraone; with imide bond), SSC-2 ((E)−2,5-dihydroxy-4-(((4-(10,15,20-triphenylporphyrin-5-yl)phenyl)imino)methyl) benzaldehyde; with imine bond) and SSC-3 (2-(4-(5,10,15-triphenylporphyrin-20-yl)

phenyl)−1H-benzo[d]imidazole-5,6-diamine; 1, 3-diazole (imidazole) bond), were synthesized on the water surface under the sequential approach, assisted by a charged surfactant monolayer, as schematically depicted in Fig. 1a. The on-water surface reaction comprises three steps in the site-selective imide bond formation from R1 (4-(5,10,15-triphenylporphyrin-20-yl)aniline) and R2 (perylene-3,4,9,10-tetracarboxylic dianhydride) towards SSR-1: (i) spreading of sodium oleyl sulfate (SOS) surfactant on the water surface in a beaker having 6 cm diameter resulting in the formation of the surfactant monolayer[43,44], (ii) after 15 minutes, an aqueous acidic solution of R1 was injected in the water subphase resulting in the pre-organization of R1 monolayer underneath the surfactant monolayer, which was facilitated by the electrostatic interaction between the protonated R1 and the anionic head group of SOS, (iii) after 1 hour, an aqueous solution of R2 was added into the water subphase leading to the diffusion towards the pre-organized monolayer of R1 as illustrated in Fig. 1b. The reaction was kept at room temperature under ambient conditions for 24 hours, presenting a macroscopic shiny brownish-orange colored film on the water surface upon visual inspection (Fig. 1c). The resulting film was collected from the water surface and analysed using matrix-assisted laser desorption/ionization–time-of-flight mass spectrometry (MALDI–TOF MS). A peak was observed at an m/z value of 1003.28 with an isotopic distribution that corresponds to the site-selective one-sided product SSC-1 (Figs. 1d, 1e), rather than the generally expected two-sided imide product that can easily be formed in bulk organic synthesis at elevated temperatures (above 100 °C) (Fig. 1a).

To examine the generality of site-selective reactions on the water surface, a similar approach was extended to other chemical reactions involving porphyrin-based molecules, i.e., an imine site-selective reaction (SSR-2) and an imidazole site-selective reaction (SSR-3). Note that for SSR-3, a cationic surfactant called cetyltrimethylammonium bromid (CTAB) was employed to facilitate the adsorption of the negatively charged R4 to the water surface. In both cases, a similar selectivity was achieved from the respective site-selective compounds SSC-2 and SSC-3 on the water surface (Fig. 1f, g). The resultant compounds were further characterized by using proton nuclear magnetic resonance spectroscopy ($^1$H NMR), attenuated total reflectance Fourier transform infrared spectroscopy (ATR-FTIR), and Raman spectroscopy (Supplementary Figs. 1–4). In contrast, the same chemical reactions (imide-, imine-, and imidazole-reactions) were performed in water under similar experimental conditions (pH, temperature, time, and concentration) without using a surfactant monolayer, and only the reagent molecules could be identified by mass spectrometry (Supplementary Fig. 5). Therefore, the overall reaction does not proceed in the bulk subphase. This control experiment suggests that the on-water surface confinement of metal-free porphyrin molecules reinforced by a surfactant monolayer is imperative for enhancing both the chemical reactivity and site-selectivity.

### In-situ structural insight into the sequential assembly of SOS and R1 on the water surface
To gain a deeper understanding of the pre-organization of reactant molecules in a well-defined structure under the sequential assembly, we performed in-situ step-by-step grazing incidence X-ray diffraction (GIXD) measurements directly on the water surface. GIXD is very sensitive towards the in-plane molecular structure of monolayers (schematically depicted in Supplementary Fig. 6), beginning with step-I, i.e., the surfactant monolayer formation. A 2D long-range ordered structure of the SOS surfactant monolayer was confirmed immediately after evaporation of the solvent (chloroform) on the water surface. The measured GIXD scattering profile (Fig. 2a) exhibited seven distinct and sharp diffraction peaks (for details, see Supplementary Tab. 1), which correspond to a unit cell with a = 5 Å, b = 7.48 Å, and γ = 89.96°, and matches well with the 2D lattice structure of the SOS surfactant monolayer derived from theoretical modeling (Fig. 2b, c and Supplementary Figs. 7 and 8).

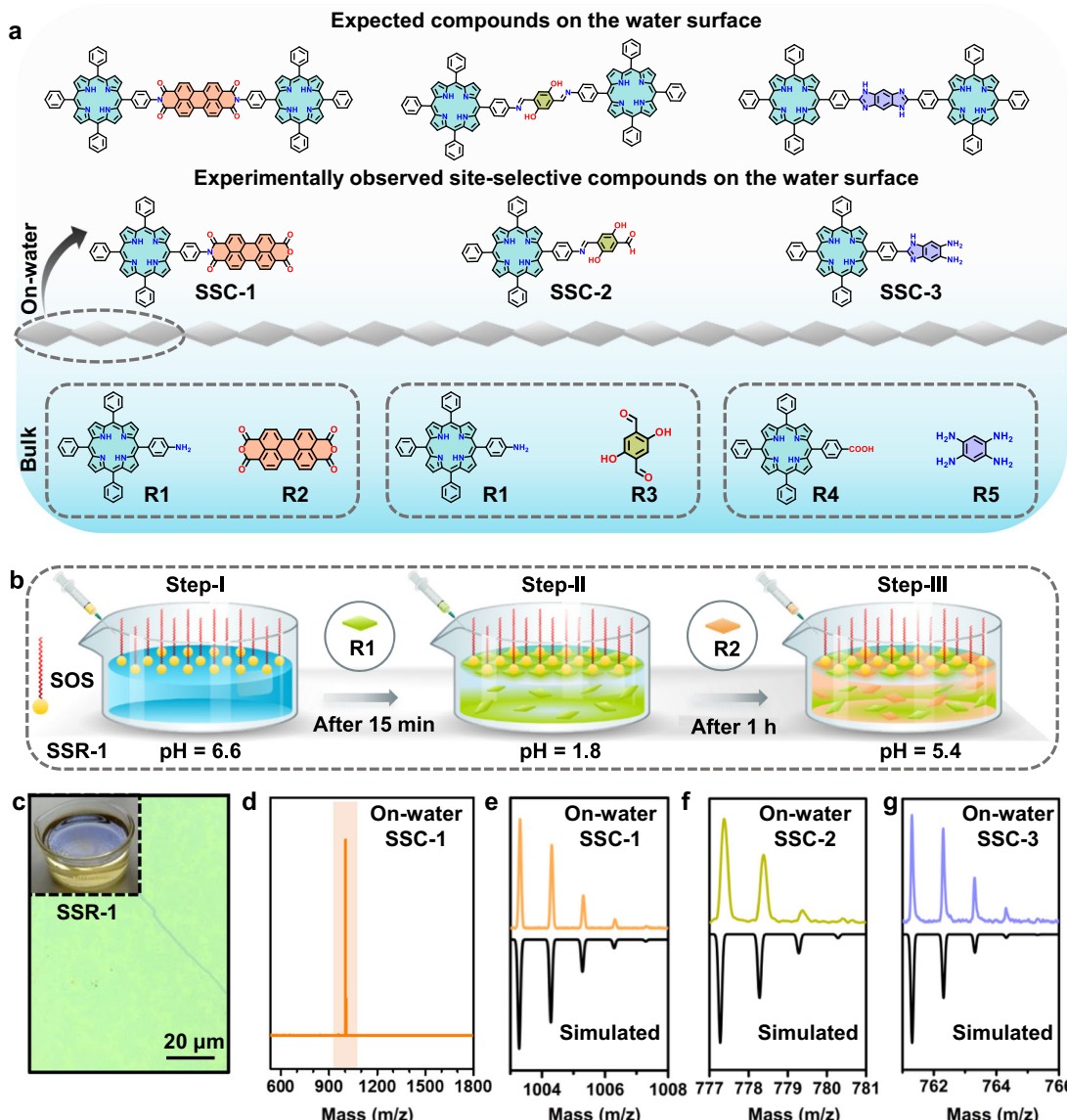

**Fig. 1 | Site-selective chemical reactions. a** Schematic of experimentally observed site-selective and expected compounds on the water surface. **b** Stepwise schematic representation of the SSR−1 on the water surface. **c** Optical microscope image of a SSC−1 film on a SiO₂/Si substrate. Inset: photograph of the SSR−1 beaker, clearly indicating a shiny film on the water surface. **d** MALDI-TOF mass spectra of SSC-1 synthesized on the water surface (highlighted in orange color). HR-MALDI-TOF mass spectra of (**e**) SSC-1, (**f**) SSC-2, and (**g**) SSC-3 synthesized on the water surface.

After achieving the crystalline structure of the SOS surfactant monolayer on the water surface, we proceeded to the next step, which involved the pre-organization of R1 underneath the surfactant monolayer, i.e., step-II (Supplementary Fig. 9). Two hours after adding R1 to the water subphase, the measured GIXD scattering profile showed distinct diffraction rings with unit cell parameters of a = 13.31 Å, b = 9.40 Å and γ = 95.40° (Fig. 2d and Supplementary Tab. 2). Furthermore, the thin film of the pre-organized R1 on the water surface was transferred onto a copper grid for selected-area electron diffraction (SAED) measurement, and the resultant SAED pattern (Fig. 2e) is in good agreement with the structure derived from the GIXD experiments (Figs. 2d-2h). The pre-organized R1 structure on the water surface exhibits significantly smaller lattice parameters than the size of individual R1 molecules (a = 16.6 Å and b = 14.8 Å) (Supplementary Fig. 10). We attribute this result to the formation of a unique 2D-confinement of R1 in J-aggregated packing guided by the surfactant monolayer, which is an energetically more favorable state (Supplementary Fig. 11).

For a further understanding of the structural relationships/interplay between the SOS crystalline surfactant monolayer and the R1 structure on the water surface, GIXD measurements were carried out with particular focus on the characteristic (110) Bragg peak of the surfactant at $Q_{xy} = 1.51$ Å⁻¹ before and after the addition of R1 in the water subphase (Fig. 2h). From the diffraction pattern, we note that the (110) Bragg peak is intact after the addition of R1 (including 2 hours of diffusion) without any change in peak position. Additionally, the (3-10) diffraction ring of R1 exactly coincides with the surfactant (110) peak, suggesting an epitaxial growth of R1 self-assembled superstructure underneath the surfactant monolayer (Figs. 2f, 2h). To analyze what kind of epitaxial relationship forms between the two lattices, we performed calculations analyzing the theoretically possible azimuthal alignments between the surfactant and R1 assembly lattices (for a detailed discussion, see Supplementary Fig. 12). From the established surfactant unit cell and the R1 lattice, the areal density ratio of surfactant molecules ($N_{SOS}$) to R1 molecules ($N_{R1}$) can be calculated directly from the inverse ratio of the respective unit cell areas: $N_{SOS}$ /

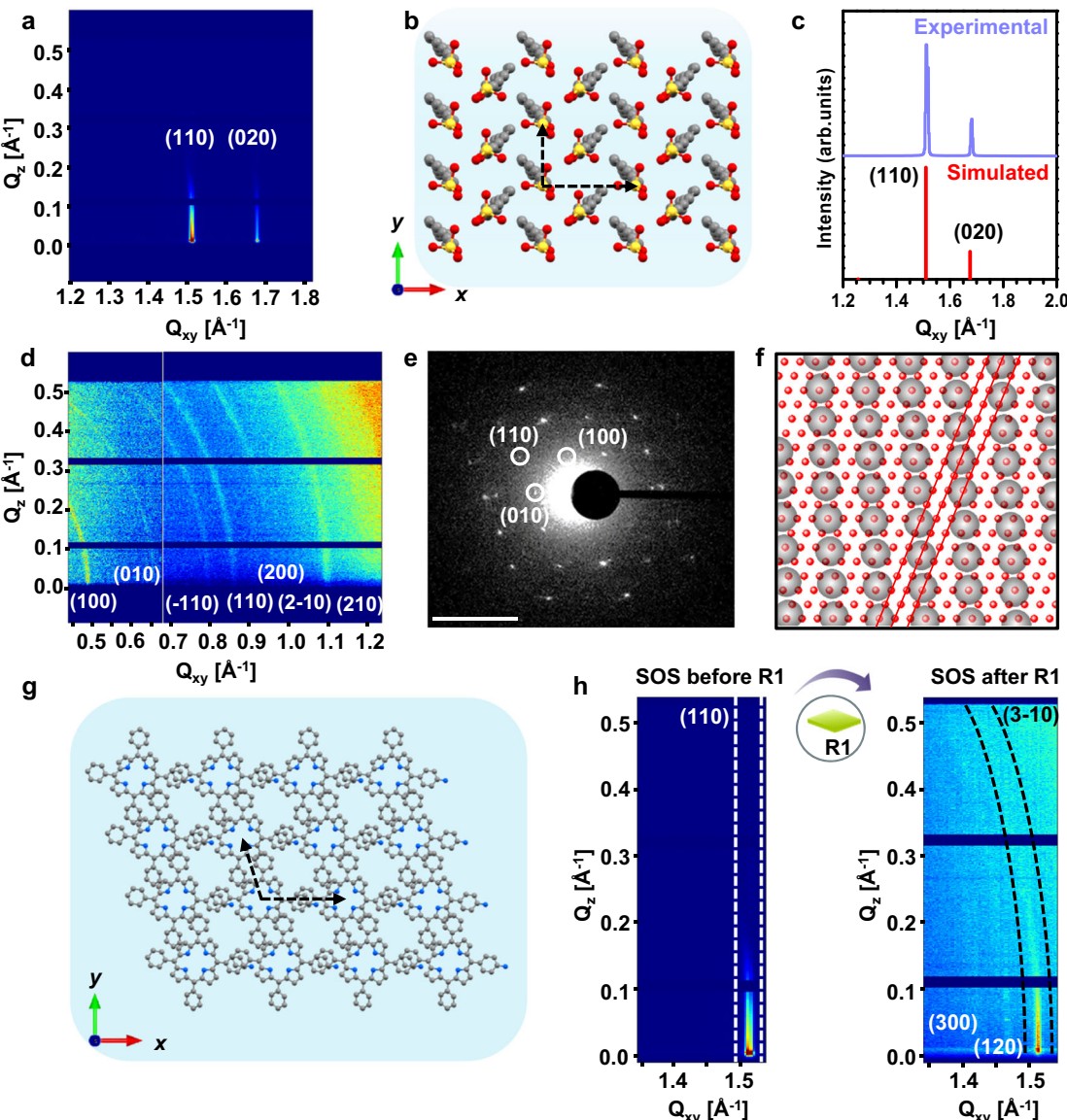

**Fig. 2 | Structural insight into the sequential assembly.** In-situ step-by-step grazing incidence X-ray diffraction (GIXD) measurements directly on the water surface. **a**, **b** Step-I, diffraction pattern and simulated structure of SOS surfactant monolayer. **c** Experimental and simulated diffraction pattern of the 2D lattice structure of the SOS surfactant monolayer. **d** Step-II diffraction pattern of the on-water reaction after injecting R1. **e** SAED pattern of the transferred R1 film: inset, scale 2 nm⁻¹. **f** Overlapping the crystal structure of the SOS surfactant monolayer and R1 shows epitaxial growth. The red lines guide the eyes to the point-on-line coincidence. **g** Lattice structure of the R1 monolayer on the water surface. **h** Characteristic (110) Bragg peak of the surfactant at $Q_{xy} = 1.51\,Å^{-1}$ before and after the addition of R1 into the water subphase.

$N_{RI} = A_{RI}/A_S = 125.2\,Å^2/18.7\,Å^2 \approx 6.7 \approx 20{:}3$ (Supplementary Fig. 13). We can conclude from this that 3 R1 molecules interact with 20 surfactant molecules.

## Confinement of J-aggregated R1 on the water surface

Among the three reaction steps mentioned above, step-II is the most critical as it facilitates the pre-organization of R1 underneath the surfactant monolayer, thus guiding the formation of a distinctive site-selective product in step-III. In order to comprehend the uniqueness of the on-water surface chemical reaction, we probed step-II in both the water surface film and water subphase with various characterization techniques (Figs. 3a-3f). (i) The UV-vis absorption spectra of R1 for the on-water surface film revealed a significant red shift in the Soret and Q-bands[45] (461 and 700 nm) compared to the water subphase (peaked at 430 and 650 nm) (Figs. 3a, 3d, and 3e). (ii) A similar red shift was observed in the photoluminescence spectra of the on-water surface

film (710 and 760 nm) compared to the water subphase (670 and 720 nm) (Fig. 3b). (iii) In the Raman spectroscopy, a noticeable red shift was observed only in the NH₂-bending mode (1620 to 1594 cm⁻¹) in the on-water surface film as compared to the water subphase (Supplementary Fig. 14). These distinctive observations suggest the formation of a J-aggregated structure[40,46] with a short slip distance in the R1 monolayer (Fig. 3e) and demonstrate the strong polarized-π interaction in the R1 structure on the water surface, which is well supported by the theoretical calculation (Fig. 3c and Supplementary Figs. 15-19). A uniform coverage of the J-aggregated binding motif was revealed at the microscopic level (scale of several microns) based on 2D-Raman microscopy (Supplementary Fig. 14). We observed a significant enhancement in the J-aggregation behavior of R1 on the water surface upon decreasing the pH (i.e., towards a more acidic nature) of the water subphase (Fig. 3f), and a similar trend was observed by the surface pressure measurements (Supplementary Fig. 20). These

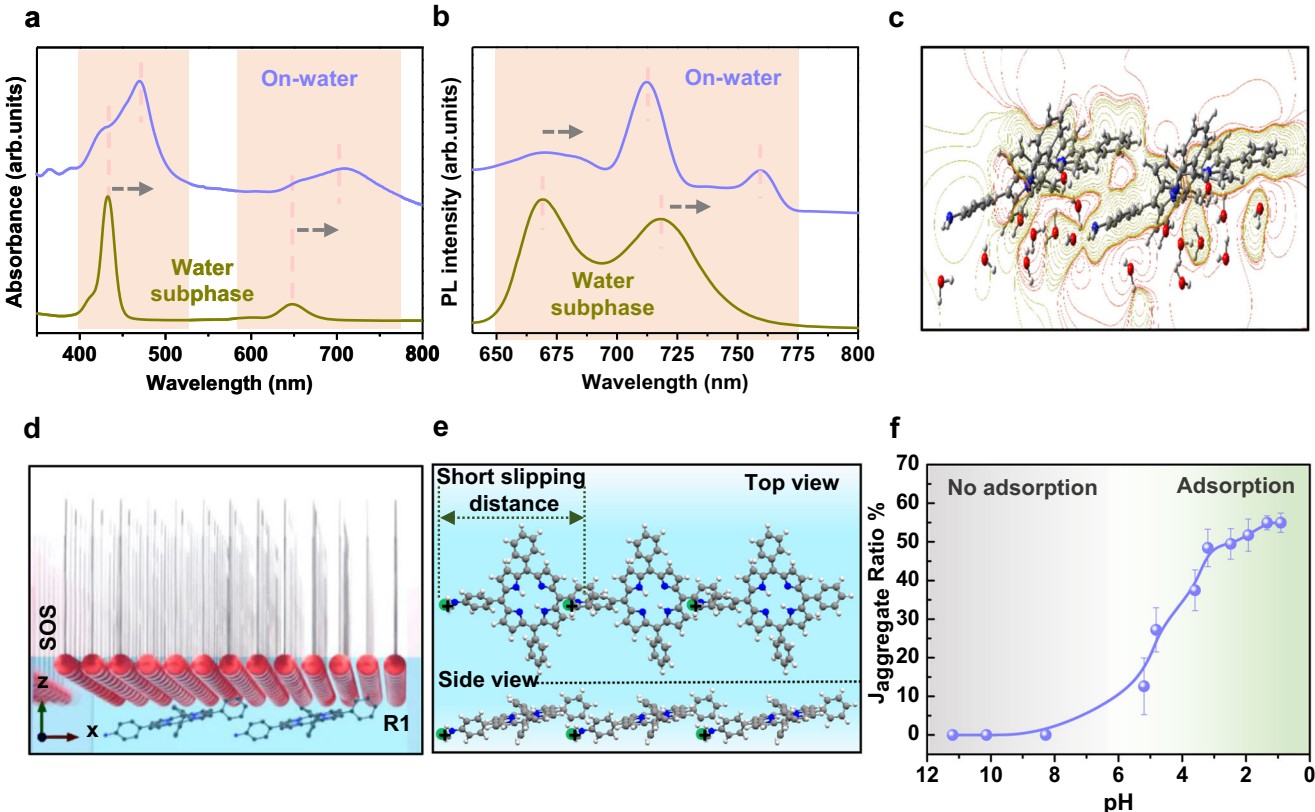

**Fig. 3 | J-aggregated R1 on the water surface. a**, **b** UV-vis and photoluminescence spectra of the water surface film and the solution of the water subphase after the addition of R1 (step-II), Soret and Q-bands of R1 highlighted by orange color. **c** The transversal view of the electrostatic potential surface (EPS) contour cut plot derived from theoretical calculations establishes the charge distribution within the R1 structure. **d** Schematic illustration of the J-aggregated R1 structure underneath the surfactant monolayer. **e** Zoomed-in image of top and side view of J-aggregated R1 monolayer indicating short-slip distance. **f** pH-dependent ratiometric UV-vis spectra of the transferred R1 film as schematically shown highlighted by gray and green color (based on the absorbance peak shift from 430 nm to 460 nm) and the error bars illustrate the standard deviation derived from three replicated measurements.

findings suggest that the acidic nature of the subphase has an influence on the formation of J-aggregates, likely through the protonation of the porphyrin molecules, and the optimal pH range for initiating R1 adsorption at the interface, which involves electrostatic interactions under the surfactant monolayer, is between pH 0.9 and 3.8.

## In-situ monitoring of sequential on-water surface chemical reactions

To elucidate the mechanism of the site-selective chemical reaction on the water surface, we monitored the time evolution of the SFG response at the SOS-water interface (step-I) and after the addition of R1 in the water subphase (step-II). SFG allows the investigation of specific molecular sub-groups and their temporal evolution during the reaction. The SFG setup is schematically depicted in (Fig. 4a) (for details, see Supplementary Figs. 21 and 22). The time evolution of the imaginary part of the SFG susceptibility (the equivalent of the surface infrared absorption spectrum) reflects the interfacial chemistry and was collected in the *ssp* polarization combination (denoting *s*-, *s*-, and *p*-polarized SFG, visible and IR, respectively) as shown in (Fig. 4b). The spectrum at time $t = 0$ (i.e., the time when we injected the R1 into the water subphase) shows a negative $2870\ cm^{-1}$ symmetric C-H stretch peak, a negative $2930\ cm^{-1}$ C-H Fermi resonance peak, and a positive $2970\ cm^{-1}$ antisymmetric C-H stretch peak[47,48]. The sign of these peaks reflects the up- or down-orientation of transition dipole moments of the vibrations. These C-H stretch features are common for the water-lipid/surfactant interface[49,50]. Furthermore, the spectrum possessed a positive O-H stretch band spanning the $3100\text{-}3600\ cm^{-1}$ region. The positive sign of the O-H stretch band indicates that the O-H groups of the interfacial water molecules are

*up*-oriented by the negatively charged head group of SOS[50–52]. After injecting R1 into the water subphase in step-II, the positive O-H stretch peak gradually decreased with time, and the signal almost vanished after the interface reached the equilibrium state (Fig. 4b). Note that the charged N-H stretch mode is generally featureless and indistinguishable from the O-H stretch[50]. We also performed an experiment in the C-N and N-H stretch mode regions, and did not observe these modes for R1. The O-H stretch peak decreased because the positively charged R1 screens the negative surface charge of the SOS head group underneath the SOS monolayer. The presence of R1 underneath the anionic SOS surfactant monolayer was further confirmed by the enhanced negative $3060\ cm^{-1}$ aromatic C-H stretch mode with time[53].

To understand the process of interfacial charge screening between the SOS and R1 assembly on the water surface, we estimated the surface charge density of the SOS surfactant monolayer on the water surface (with a subphase pH of 1.8) using the Gouy-Chapman model[50,54]. The estimated value was $-0.037 \pm 0.002\ C/m^2$, which is lower than the value of the SOS surfactant monolayer on the water surface (with a subphase pH of 6.6). The lower surface charge density is expected for an acidic subphase given the sulfate group's pKa value of $2.0$[55] (Supplementary Fig. 22). The surface charge appeared fully screened already during the equilibrium process (~35 minutes after adding R1 to the water subphase). The trend of the intensity change of the $Im\chi^{(2)}$ spectra (Fig. 4b) indicates the total surface charge of the SOS-R1 co-assembled structure (Fig. 4c).

After reaching equilibrium, we performed atomic force microscopy (AFM) measurements of the on-water surface film (transferred onto $SiO_2/Si$ substrate by the Langmuir-Schaefer method) to examine

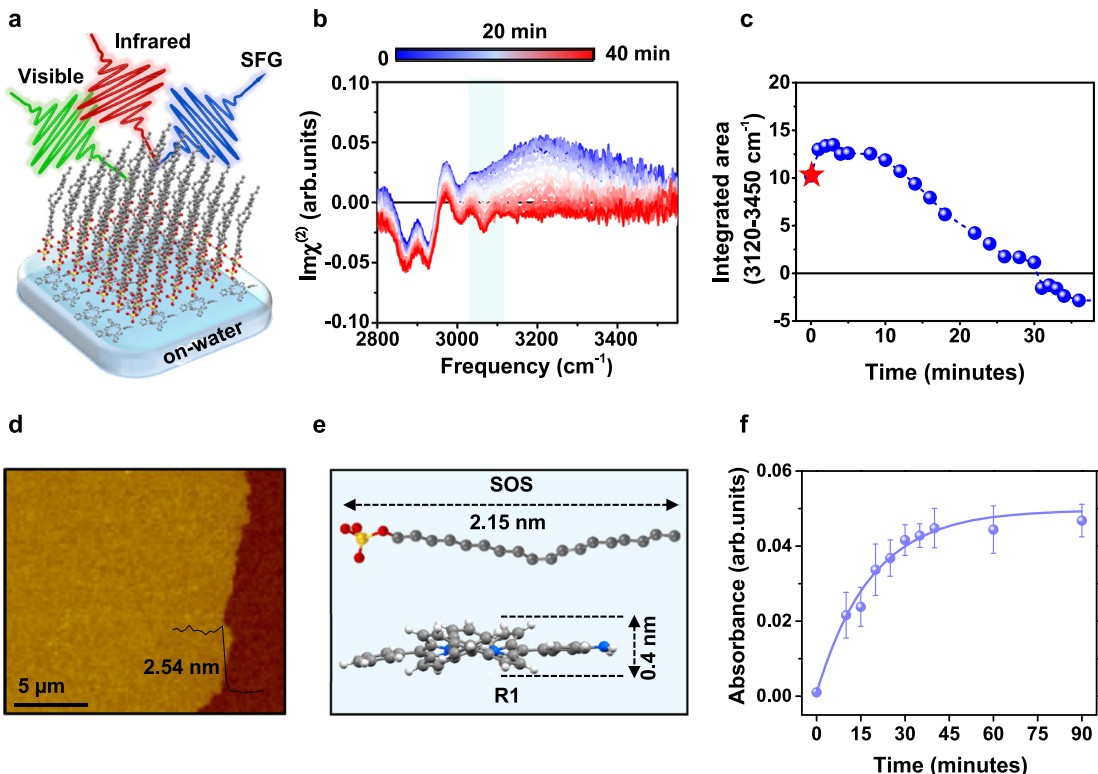

**Fig. 4 | Real-time monitoring of sequential chemical reactions on the water surface.** Time evolution of SFG spectra for the R1 adsorption underneath the surfactant monolayer. **a** The schematic illustration of the SFG measurement. **b** The time evolution of the imaginary part of the SFG susceptibility at the SOS-water interface (step-I) after adding R1 into the water subphase (3060 cm⁻¹ aromatic C-H stretch mode highlighted by light green color). **c** The integrated area of O-H stretch mode (integrated range is from 3120 to 3450 cm⁻¹, indicating the change in the number of R1 layers adsorbed underneath the SOS surfactant hydrogen-bonded water molecules) as a function of time (the red star corresponds to the charge density of 0.037 ± 0.002 C/m²). **d** AFM image showing the collective contribution from the adsorbed R1 monolayer and SOS surfactant monolayer. **e** The thickness of the monolayer SOS surfactant is 2.15 nm, whereas the monolayer R1 has a thickness of 0.4 nm. **f** Time-dependent UV-vis absorbance study after adding R1 into the water subphase (absorbance peak at 460 nm) and the error bars illustrate the standard deviation derived from three replicated measurements.

the number of R1 layers adsorbed underneath the SOS surfactant monolayer. The measured thickness of SOS-R1 assembly was 2.54 nm, which closely matches the collective contribution of monolayer thicknesses of SOS (2.1 nm) and R1 (0.4 nm). This result points to the adsorption of a single layer R1 underneath the SOS surfactant monolayer (Figs. 4d, 4e). We also observed similar R1 adsorption behavior in time-dependent UV-vis spectroscopy, and saturation of R1 absorbance peak at 460 nm was reached after 30 minutes (Fig. 4f), in line with the results of the in-situ SFG studies. Additionally, the importance of surface charge was realized by performing SFG measurements in the absence of SOS monolayer showing no R1 adsorption on the water surface in time-dependent Imχ⁽²⁾ spectra (Supplementary Fig. 23).

Next, we explored Step-III, where amino-substituted porphyrin R1 reacts with perylenetetracarboxylic dianhydride R2 on the water surface, by in-situ GIXD study to understand the role of J-aggregation towards site-selectivity. The azimuthal alignment epitaxy of the SOS-R1 assembly in step-II remained intact on the water surface after the addition of R2 in step-III, suggesting the retention of the J-aggregated packing structure of R1, as evidenced by the time-dependent GIXD measurements (Supplementary Fig. 24). The energetically favorable epitaxial assembly of SOS-R1 limits the movement or reconstruction of the J-aggregated structure of R1, which directs the precise orientation of perylenetetracarboxylic dianhydride R2 with respect to the amino functional group of R1, thereby, promoting the selective formation of a one-sided imide bond over a two-sided imide product.

## Dynamic interface to favor multilayer site-selective products

We noted that the above sequential assembly approach enabled the further formation of multilayer site-selective product in step-III, which could be elucidated by using in-situ SFG spectroscopic technique. After completing step-II, R2 was added to the water subphase, and the time evolution of the chemical reaction was monitored (as shown in Figs. 1b, 4c, and 5a). It is notable that with increasing waiting time (15 minutes), the aromatic C-H peak decreases, and the O-H stretch band appears again in the SFG Imχ⁽²⁾ spectra (Fig. 5a). The reappearance of the positive O-H stretch band indicates the negatively charged interface, which can be attributed to the release of $H_3O^+$ into the water subphase during the chemical reaction between R1 and R2. Consequently, the total charge at the interface becomes negative due to the release of proton to the bulk solution through the chemical reaction (Fig. 5b–d). The SOS surfactant head group's unscreened negative charge and the well-defined template structure act as a driving force for the further adsorption of positively charged R1 from the water subphase. The continuous imide bond formation leads to charge release, followed by further R1 adsorption, which eventually allows layer-by-layer growth of multilayer site-selective product (SSC-1) that reaches equilibrium after 12 hours with a thickness of ~13 nm, as demonstrated by the time-dependent AFM analysis (Fig. 5c and Supplementary Fig. 25). Similarly, the intensity of C-N stretching and C = O stretching mode was observed to increase in a time-dependent SFG study after adding R2 into the water subphase, indicating the formation of a multilayer site-selective product (Supplementary Fig. 26). The layer-by-layer growth of multilayer site-selective imide product on the water surface discussed above serves as a specific example illustrated in Fig. 5d. In addition to the site-selective imide reaction, we have also discovered that this sequential assembly approach on the water surface can also be applied to other chemical reactions, such as imine and 1, 3-diazole (imidazole),

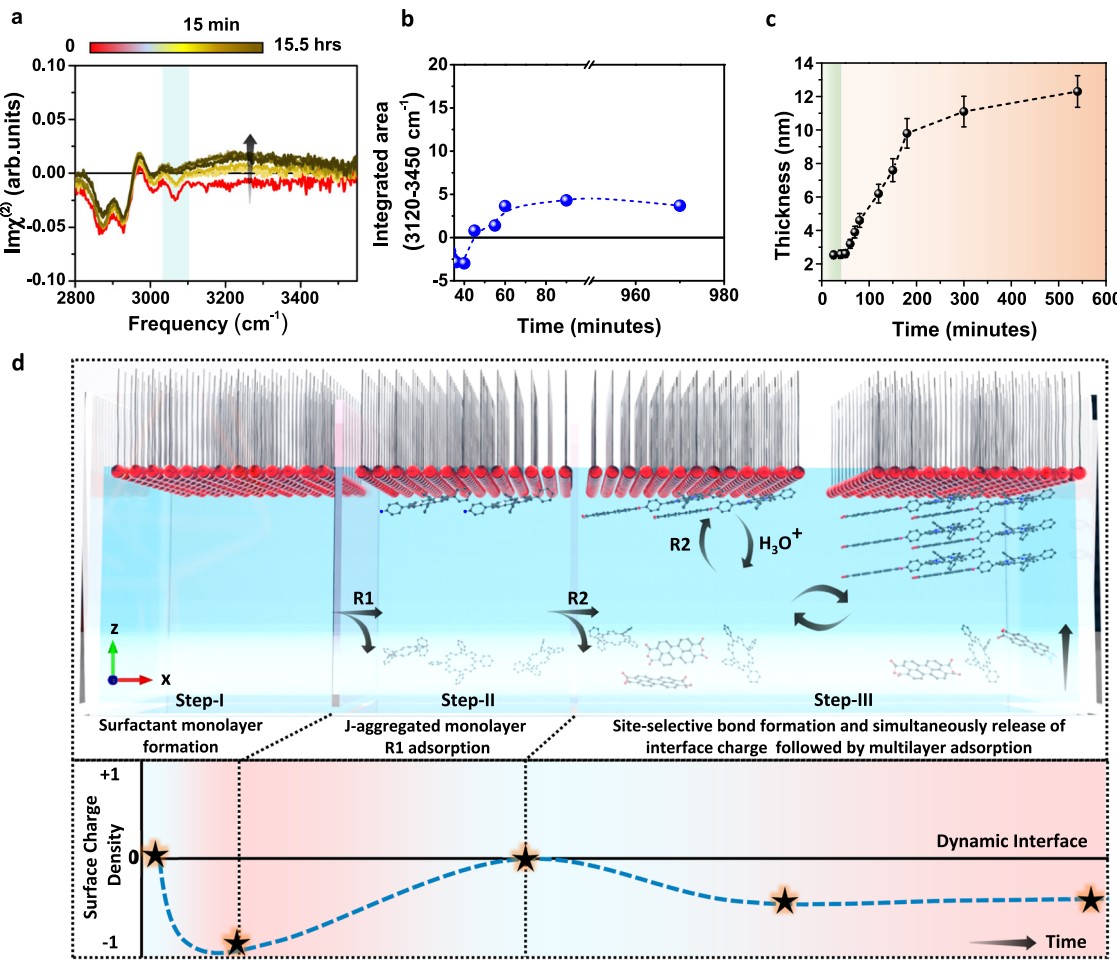

**Fig. 5 | Multilayer site-selective products. a** Time evolution (3060 cm⁻¹ aromatic C-H stretch mode highlighted by light green color). and (**b**) the integrated area of the SFG spectra after adding R2 into the water subphase along with the time evolution (integrated range is from 3120 to 3450 cm⁻¹, indicating the change in hydrogen-bonded water molecules). **c** AFM analysis of the film thickness as a function of time after adding R2 into the water subphase (monolayer and multilayer adsorption highlighted by green and orange color, respectively) and the error bars illustrate the standard deviation derived from three replicated measurements. **d** A schematic illustration of the dynamic interface showing the step-wise mechanism for the multilayer growth of the site-selective product on the water surface.

resulting in the exclusive, multilayer growth of one-sided site-selective products.

## Discussion

The sequential assembly approach for site-selective chemical reactions has two basic reagent requirements: (i) reactants with pH-dependent charges and (ii) surfactants and reagents with opposite charges. In all site-selective reactions, namely SSR-1, SSR-2, and SSR-3, the self-assembly and confinement of porphyrin molecules on the water surface are governed by J-aggregation. The presence of charged components facilitates electrostatic interactions with oppositely charged surfactants, while the J-aggregated confinement enhances the reactivity of the targeted site, enabling selective bond formation (Supplementary Fig. 27). This approach holds promise for various types of selective chemical reactions, including chemo-, diastereo-, and enantio-selectivity.

In summary, we have demonstrated a sequential assembly approach for achieving site-selective chemical reactions on the water surface, using a J-aggregated packing structure of porphyrin molecules. We have successfully applied this approach to a wide range of reversible and irreversible chemical reactions, including imide-, imine-, and 1, 3-diazole (imidazole)- bond formation, with precise control over site-selectivity. Through a combination of in-situ techniques and theoretical calculations, we have elucidated a step-wise molecular level

mechanism, which highlights the crucial role of the charged surfactant monolayer in directing the pre-organization of reactant molecules in an epitaxial manner. This pre-organization, facilitated by the restricted geometry of J-aggregation, drives the site-selectivity on the water surface with a controlled multilayer growth. Our findings not only shed light on the fundamental understanding of on-water surface chemistry, but also have significant implications for the fields of synthetic, biological, and materials chemistry, as the sequential assembly approach holds great potential for designing functional materials and biomimetic systems.

## Methods

### General characterization

¹H NMR spectra were recorded at room temperature with a Bruker Avance III HD 300 spectrometer operating at 300 MHz. DMSO-d₆ (DMSO, dimethylsulfoxide) was used as the solvent. The high-resolution matrix-assisted laser desorption/ionization time-of-flight (MALDI-TOF) mass spectrometry was performed on a Bruker Autoflex Speed MALDI-TOF MS using trans-2-[3-(4-tert-butylphenyl)−2-methyl-2-propenylidene] malononitrile (DCTB) or dithranol as matrix. Optical microscopy (Zeiss), AFM (Bruker Multimode 8 HR), and TEM (Zeiss, Libra 200 KV) were used to investigate the morphology and structure of the samples. Thin films were deposited on a Si substrate for SEM measurements and on copper grids for TEM measurements. All the

optical microscopy and AFM images were recorded on a 300 nm $SiO_2$/Si substrate. UV–vis absorption spectra were recorded on a UV–vis–NIR spectrophotometer Cary 5000 device at room temperature on a quartz glass substrate. Photoluminescence spectra were measured on the PerkinElmer fluorescence spectrometer LS 55. ATR-FTIR was performed on a Tensor II system (Bruker) with an attenuated total reflection unit, and the samples were prepared by depositing the thin films onto a copper foil. Time-dependent surface-pressure measurements were carried out by the Langmuir-Blodgett trough (KSV NIMA, Finland). The trough was equipped with a platinum Wilhelmy plate, a Teflon dipper, and a pair of Delrin barriers.

### In-situ on-water surface GIXD

In-situ Grazing Incidence X-ray Diffraction (GIXD) measurements were performed at the SIRUS beamline at SOLEIL, St. Aubin, France. The energy of the beam was 8 keV, the spot size was 2000 μm x 150 μm and the incidence angle was 2 mrad. The measured films were grown on the air-water interface in either a Langmuir trough with adjustable barriers or a PTFE trough with a diameter of 10 cm and a depth of 2 mm, which were placed in a helium filled enclosure to reduce air scattering and beam damage to the film. A Dectris Pilatus 1 M area detector with Soller slits (0.06° resolution) was scanned along the in-plane, horizontal angle (2θ) to record images (exposure times 5 – 10 s). The individual images were horizontally integrated to obtain the vertical intensity distribution ($I(Q_z)$) at each 2θ angle. These 1D spectra were then combined to create a 2D- Qxy-Qz intensity map that was analyzed using WxDiff. The setup was calibrated using a film of Behenic acid at the water surface.

### In-situ on-water surface Sum Frequency Generation (SFG) spectroscopy

The femtosecond Ti: Sapphire amplified laser system (Spitfire Ace, Spectra-Physics, ~800 nm, ~40 fs, 1 kHz) generates 5 W output power. We use 2 W to pump an optical parametric amplifier (TOPAS, light conversion) with a non-collinear difference frequency generation stage to generate broadband infrared (IR) pulses. Another 1 W of the laser output passes through an etalon or pulse shaper to generate narrowband visible pulses ( ~10 cm$^{-1}$). The IR and visible beam are firstly focused onto a 20 μm-thick *y*-cut quartz plate as the local oscillator. Then these beams collinearly pass through a 2 mm $SrTiO_3$ plate for the phase modulation and are focused on the sample surface at angles of incidence of 45°. The visible and IR powers were 10 μJ and 2 μJ, respectively. The SFG signal from the sample interferes with the SFG signal from the localoscillator (LO), generating the interference fringe. The interference fringe is measured by dispersing the signal in a spectrometer (Princeton Instruments) and detected by a liquid nitrogen-cooled CCD camera (Princeton Instruments). The complex spectra of second-order nonlinear susceptibility ($\chi^{(2)}_{eff}$) are obtained via the Fourier analysis of the interferogram and normalization by a *z*-cut quartz (*zqz*) crystal. For the homodyne measurement in low frequency region, visible (13 μJ) and IR (5 μJ) incident angles were 36° and 41°, respectively. The generated SFG pulse was subsequently focused onto a spectrograph (Acton SP 300i, Princeton Instruments), and detected with an EM-CCD (Newton, Andor Technology).

### Theoretical calculations

We use density functional theory (DFT) techniques to calculate the electronic properties of the porphyrin system studied in this article. We report the results using the wB97XD functional, and the 6-311 + G(d) basis set as implemented in the Gaussian16 code. TD-DFT was used for the calculation of the linear excitations. We use water for the solvent calculations within the Polarizable Continuum Model (PCM) using the integral equation formalism variant (IEFPCM), as implemented in Gaussian16 (revision number: C.01).

### Site-selective reaction, SSR-1

Synthesis of site-selective compound SSC-1. Milli-Q water (40 ml) was injected into a beaker (80 ml, diameter = 6 cm) to form a static air-water interface. Then, 10 μl of SOS (1 mgml$^{-1}$ in chloroform) was spread onto the surface. The solvent was allowed to evaporate for 10 minutes, and then R1 (100 μl, 1 mgml$^{-1}$ in 0.12 M HCl solution) was gently added to the water subphase using a syringe. After 1 hour, R2 (150 μl, 1 mgml$^{-1}$ in 0.04 M LiOH aqueous solution) was gradually added to the water subphase with a syringe. The reaction was then kept undisturbed at room temperature for 24 hours. Upon visual inspection, the reaction results in the formation of a brownish-orange film on the surface of the water. The resulting film was collected from the water surface onto the substrate by the horizontal dipping method.

### Site-selective reaction, SSR-2

Synthesis of site-selective compound SSC-2. Milli-Q water (40 ml) was injected into a beaker (80 ml, diameter = 6 cm) to form a static air-water interface. Then, 10 μl of SOS (1 mgml$^{-1}$ in chloroform) was spread onto the surface. The solvent was allowed to evaporate for 10 minutes, and then R1 (100 μl, 1 mgml$^{-1}$ in 0.15 M HCl solution) was gently added to the water subphase using a syringe. After 1 hour, R3 (120 μl, 1 mgml$^{-1}$ in aqueous solution) was gradually added to the water subphase with a syringe. The reaction was then kept undisturbed at room temperature for 24 hours. Upon visual inspection, the reaction results in the formation of a brownish film on the surface of the water. The resulting film was collected from the water surface onto the substrate by the horizontal dipping method.

### Site-selective reaction, SSR-3

Synthesis of site-selective compound SSC-3. Milli-Q water (40 ml) was injected into a beaker (80 ml, diameter = 6 cm) to form a static air-water interface. Then, 15 μl of CTAB (1 mgml$^{-1}$ in chloroform) was spread onto the surface. The solvent was allowed to evaporate for 10 minutes, and then R4 (120 μl, 1 mgml$^{-1}$ in 0.05 M LiOH aqueous solution) was gently added to the water subphase using a syringe. After 1 hour, R5 (140 μl, 1 mgml$^{-1}$ in 0.30 M HCl solution) was gradually added to the water subphase with a syringe. The reaction was then kept undisturbed at room temperature for 24 hours. Upon visual inspection, the reaction results in the formation of a purple film on the surface of the water. The resulting film was collected from the water surface onto the substrate by the horizontal dipping method.

## Data availability

The data that support the plots within this paper and other findings of this study are available from the corresponding author upon request.

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

## Acknowledgements

This work was financially supported by the EU Graphene Flagship (GrapheneCore3, no. 881603), an ERC starting grant (FC2DMOF, grant no. 852909), an ERC Consolidator Grant (T2DCP), H2020-MSCA-ITN (ULTIMATE, no. 813036), H2020-FETOPEN (PROGENY, 899205), CRC 1415 (Chemistry of Synthetic Two-Dimensional Materials, no. 417590517), SPP 2244 (2DMP), GRK2861 (No. 491865171), as well as the German Science Council and Center of Advancing Electronics Dresden. R.D. thanks Taishan Scholars Program of Shandong Province (tsqn201909047) and National Natural Science Foundation of China (22272092). We thank Chun-Chieh Yu and Kuo-Yang Chiang for the help during the SFG measurements at MPIP Mainz. We gratefully acknowledge support from the MaxWater initiative from the Max Plank Society. We acknowledge SOLEIL for provision of synchrotron radiation facilities and we would like to thank Arnaud Hemmerle for assistance in using beamline SIRIUS. A.P. thanks Dr. Chandrasekhar Naisa for the NMR measurements. The authors acknowledge the Center of Advancing Electronics Dresden, the Dresden Center for Nanoanalysis at TUD, and P. Formanek and A. Fery for the use of the TEM facility at IPF.

## Author contributions

X.F. and R.D. conceived and designed the project. A. P. planned the experimental sessions, synthesized, and characterized all site-selective compounds. A.P., X.Y., T.S., and Y.N. conducted the SFG experiments; X.Y., T.S., Y.N., and M.B. analyzed the SFG data. A.P., R.D., M.H., P.F., and S.M. contributed to the in-situ GIXD experiments; M.H. and S.M. analyzed the in-situ GIXD data. D.B., A.Z., A.D., A.C., and G.C. contributed to the theory calculations and analysis. K.L. performed TEM imaging. N.N.N. performed AFM and Raman analysis. A.P. and X.F. co-wrote the manuscript with contributions from all the authors. All the authors discussed the results and commented on the manuscript.

## Funding

## Competing interests

The authors declare no competing interests.
