## [Peer Review File · Nature Communications]

Site-selective chemical reactions by on-water surface sequential assemblyReviewers' Comments:

Reviewer #1:

Remarks to the Author:

The authors discuss an interesting approach for site-selective chemical reactions, at the air-water interface. First they create a surfactant monolayer at the air/water interface. They observed that in case of a mono-functionalized porphyrine derivative, the porphyrin organizes underneath the surfactant monolayer, in a J-type configuration (probably one-molecule thick). Upon adding reagent 2, only one reaction product was formed in a R1-R2 1:1 ratio, in contrast to the expected 2:1 ratio, attributed to the "site-selectivity" generated by the specific environment.

It is an interesting approach. With an impressive set of experimental and theoretical approaches, the authors have acquired insight in the structure and the dynamics at play. I applaud this detailed investigation.

I'm left though with a number of questions/remarks.

- 1) It is not clear in the manuscript why the reaction is actually one-sided. Can the authors illustrate with a cartoon the reason why the other side of R1 is blocked?
- 2) What limits the thickness of the product layer underneath the surfactant layer?
- 3) While the authors present their approach as a general strategy, it is questionable how predictive the approach can become. In all fairness, this is clearly a proof-of-concept and much more research will be required to prove the generality of the approach.
- 4) I invite the authors to comment on upscaling. How much material is formed in their container with a diameter of 6 cm?

Reviewer #2:

Remarks to the Author:

A. Prasoon and coworkers report on site-selective reactions (formation of imides, imines, and imidazoles) of amino-substituted porphyrins on the water-air interface using a highly innovative programmable assembly approach. It is shown that a surfactant layer at the interface induces formation of an ordered J-aggregated structure of the porphyrins. As a result of this well-defined geometry at the interface, the porphyrin molecules react only with one of the two possible sites of the reagents. The structure at the interfaces is analyzed in detail with an impressive range of surface-sensitive experimental techniques, including SFG and GIXD, indicating multilayer growth of the site-selective products. The study under review represents an important milestone in the development of on-liquid chemistry by showing how high selectivity can be achieved by controlling the interface structure using suitable surfactants, and is a groundbreaking model case for the application of surface-sensitive techniques for the analysis of chemical reactions at liquid surfaces. The conclusions are well-supported by the data, which are of high quality. The manuscript is well-written and the illustrations are instructive and appealing. Although I have studied the manuscript and the SI very carefully, I do not see any issues that require revision. Therefore, I highly recommend publication of this remarkable manuscript without delay.

Reviewer #3:

Remarks to the Author:

In this paper, Prasoon et al. report on a site-selective chemical reactions by on-water surface programmable assembly. The authors claim that they successfully applied this approach to a wide range of reversible and irreversible chemical reactions, including imide-, imine-, and 1, 3-diazole (imidazole)- bond formation, with precise control over site-selectivity. They used in-situ techniques and theoretical calculations to identify a possible step-wise molecular level mechanism. I have some comments concerning this paper.

- 1) It is not clear what the "programmable assembly" is. They authors didn't show they can trigger specific assemblies. To me it seems that they claim to be able to trigger three reactions leading to SSC-SSC-2 and SSC-3.
- 2) Most of the used techniques focus on characterizing R1. They should instead focus on characterizing the resulting compounds SSc-1, SSC-2, SSC-3
- 3) It is not clear how Fig.5c has been produced. Where are the AFM images?
- 4) Most of the data interpretations are speculative, other interpretations are possible.
- 5) Overall this is more a technical paper than a scientific paper.

I suggest submitting this paper in a more specialised journal.

Detailed response to the comments from the reviewers

Reviewer #1 (Remarks to the Author):

General comment: *The authors discuss an interesting approach for site-selective chemical reactions, at the air-water interface. First they create a surfactant monolayer at the air/water interface. They observed that in case of a mono-functionalized porphyrin derivative, the porphyrin organizes underneath the surfactant monolayer, in a J-type configuration (probably one-molecule thick). Upon adding reagent 2, only one reaction product was formed in a R1-R2 1:1 ratio, in contrast to the expected 2:1 ratio, attributed to the “site-selectivity” generated by the specific environment. It is an interesting approach. With an impressive set of experimental and theoretical approaches, the authors have acquired insight in the structure and the dynamics at play. I applaud this detailed investigation.*

Response: We greatly appreciate the Reviewer for the highly positive and insightful comments guiding the revision of our manuscript.

I'm left though with a number of questions/remarks.

Comment 1: *It is not clear in the manuscript why the reaction is actually one-sided. Can the authors illustrate with a cartoon the reason why the other side of R1 is blocked?*

Response: We highly appreciate the valuable comment from the reviewer. In response to the suggestion, we have included a schematic cartoon illustrating a site-selective chemical reaction.

Figure R1 | A schematic illustrating a site-selective chemical reaction on the water surface.

The surfactant monolayer on the water surface guides the epitaxial assembly of R1 molecules into a well-defined J-aggregated structure and demonstrates strong polarized- π interaction within the R1 structure, thereby restricting the movement or rotation of the J-aggregated R1. This constrained geometry of the porphyrin molecules facilitates the subsequent directional alignment of the R2 reagent, leading to the selective formation of a one-sided imide product on the water surface.

ACTION:

Supplementary Information: Figure R1 is incorporated into modified SI as Supplementary Fig. 22, and the corresponding discussions are included in page SI 37.

Comment 2: *What limits the thickness of the product layer underneath the surfactant layer?*

Response: The thickness of the site-selective product layer underneath the surfactant monolayer on the water surface is influenced by various factors. Primarily, the surface charge density of the surfactant plays a crucial role on the water surface, which is likely to follow a double-layer model (Dreier et al., *Sci. Adv.* **4**, eaap7415 (2018)). In addition, the surface activity of the surfactant, the packing density, and the nature of interactions between the surfactant and the reacting molecules contribute to the determining factor of the thickness.

Comment 3: *While the authors present their approach as a general strategy, it is questionable how predictive the approach can become. In all fairness, this is clearly a proof-of-concept and much more research will be required to prove the generality of the approach.*

Response: We highly appreciate the valuable comments from the reviewer. The programmable assembly approach for site-selective chemical reactions has two basic requirements of reagents: (i) Porphyrin with pH-dependent charges and (ii) surfactants and porphyrin reagents with opposite charges. The presence of a charged porphyrin reactant facilitates electrostatic interactions with oppositely charged surfactants, leading to the self-assembly and confinement of porphyrin molecules into a J-aggregated structure on the water surface. The resultant J-aggregated confinement enhances the reactivity of the targeted site, enabling the selective bond formation. We fully agree with the reviewer that more research will be done to prove the generality of the approach. In this work, we only demonstrate the discovery of site-selective chemical reactions (imide, imine, and 1, 3-diazole (imidazole)) on the water surface, showing the proof-of-concept. We are currently exploring on-water surface site-selectivity and reactivity in a wide range of diversified reactions at ambient room temperature, such as imide, imine,

amide, azole, pyridinium, and anhydride reactions, comprising more than 13 reactions, which will be published in a separate work in the near future.

Comment 4: *I invite the authors to comment on upscaling. How much material is formed in their container with a diameter of 6 cm?*

Response: We appreciate the reviewer for providing insightful comments on upscaling. In a beaker with a diameter of 6 cm, the site-selective products form on a sub-mg scale on the water surface after 12 hours, with a thickness of ~13 nm. We are currently extending the strategy of on-water surface programmable assembly to the bulk synthesis of the desired site-selective product by designing self-assembled supramolecular micelle structures in the aqueous phase while maintaining on-water surface chemistry. With this approach, we will be able to upscale the bulk synthesis of desired site-selective products. As a preliminary result, we have achieved a yield of 92% with exceptionally high selectivity (>99%) in a wide range of diversified reactions, including imide, imine, amide, azole, pyridinium, and anhydride reactions. This result will be published in a separate report at the later stage.

Reviewer #2 (Remarks to the Author):

General comment: *A. Prasoon and coworkers report on site-selective reactions (formation of imides, imines, and imidazoles) of amino-substituted porphyrins on the water-air interface using a highly innovative programmable assembly approach. It is shown that a surfactant layer at the interface induces formation of an ordered J-aggregated structure of the porphyrins. As a result of this well-defined geometry at the interface, the porphyrin molecules react only with one of the two possible sites of the reagents. The structure at the interfaces is analyzed in detail with an impressive range of surface-sensitive experimental techniques, including SFG and GIXD, indicating multilayer growth of the site-selective products. The study under review represents an important milestone in the development of on-liquid chemistry by showing how high selectivity can be achieved by controlling the interface structure using suitable surfactants, and is a groundbreaking model case for the application of surface-sensitive techniques for the analysis of chemical reactions at liquid surfaces. The conclusions are well-supported by the data, which are of high quality. The manuscript is well-written and the illustrations are instructive and appealing. Although I have studied the manuscript and the SI very carefully, I do not see any issues that require revision. Therefore, I highly recommend publication of this remarkable manuscript without delay.*

Response: We greatly appreciate the Reviewer for the highly positive comments with recommendation for publication.

Reviewer #3 (Remarks to the Author):

General comment: *In this paper, Prasoon et al. report on a site-selective chemical reactions by on-water surface programmable assembly. The authors claim that they successfully applied this approach to a wide range of reversible and irreversible chemical reactions, including imide-, imine-, and 1, 3-diazole (imidazole)- bond formation, with precise control over site-selectivity. They used in-situ techniques and theoretical calculations to identify a possible step-wise molecular level mechanism. I have some comments concerning this paper.*

Response: We highly appreciate the valuable comment from the reviewer guiding the revision of our manuscript.

Comment 1: *It is not clear what the “programmable assembly” is. They authors didn’t show they can trigger specific assemblies. To me it seems that they claim to be able to trigger three reactions leading to SSC-SSC-2 and SSC-3.*

Response: We highly appreciate the valuable comments from the reviewer. The programmable assembly approach is defined as a step-by-step assembly process. In step I, a charged surfactant monolayer forms with a close packing structure on the water surface. In step II, the surfactant monolayer on the water surface guides the electrostatically driven, epitaxial, and aligned assembly of reagent amino-substituted porphyrin molecules, resulting into a well-defined, unique J-aggregated structure. In step III, the constrained geometry of the porphyrin molecules enables the subsequent directional alignment of the perylenetetracarboxylic dianhydride reagent, leading to the selective formation of a one-sided imide product on the water surface. It

is important to follow all these step-by-step reactions to confer site-selective chemical reactions.

Comment 2: *Most of the used techniques focus on characterizing R1. They should instead focus on characterizing the resulting compounds SSC-1, SSC-2, SSC-3*

Response: We highly appreciate the valuable comments from the reviewer. To comprehend the unique site-selectivity and reactivity on the water surface, we studied step-by-step chemical reactions in detail via various in-situ techniques such as sum frequency generation (SFG) spectroscopy and grazing incidence X-ray diffraction (GIXD), starting from step-I to step-III, including the final resulting compound, referring to the sections in the main text (*in-situ monitoring of programmable on-water surface chemical reactions* on page no. 13 and *dynamic interface to favor multilayer site-selective products* on pages no. 13-15). Among the three reaction steps mentioned above, step-II (which is related to R1) is the most important step as it facilitates the pre-organization of R1 underneath the surfactant monolayer, thus guiding the formation of a distinctive site-selective product in step-III. Site-selective compounds SSC-1, SSC-2, and SSC-3 are the final resulting compounds, and we employed ex-situ techniques, specifically proton nuclear magnetic resonance spectroscopy (¹H NMR), attenuated total reflectance Fourier-transform infrared spectroscopy (ATR-FTIR), and Raman spectroscopy, to characterize the site-selective compounds SSC-1, SSC-2, and SSC-3.

Comment 3: *It is not clear how Fig.5c has been produced. Where are the AFM images?*

Response: In response to the suggestion, we have included time-dependent AFM analysis after adding R2 into the water subphase.

Figure R3 | Time-dependent AFM analysis after adding R2 into the water subphase.

ACTION:

Supplementary Information: Figure R3 is incorporated into modified SI as Supplementary Fig. 20 in page SI 35.

Comment 4: *Most of the data interpretations are speculative, other interpretations are possible.*

Response: We highly appreciate the valuable comments from the reviewer. However, we are not clear on which data interpretation the reviewer is referring to.

Comment 5: *Overall this is more a technical paper than a scientific paper.*

Response: We highly appreciate the valuable comments from the reviewer. However, we respectfully hold a different perspective from the reviewer. We would like to also kindly share the insightful and very positive comments provided by both reviewer #1 and reviewer #2 about the important scientific discovery of the current work.

Our discovery highlights the fundamental understanding of on-water surface chemistry and also has significant implications for other fields. The key fundamental features of our study lie in:

1. Unique site-selectivity and reactivity in chemical reactions on the water surface. We discover here the site-selective chemical reactions on the water surface for the first time, which include both reversible and irreversible reactions through a programmable approach.

2. Surfactant-driven J-aggregated molecular confinement. Here, we present a novel strategy for controlling the orientation of porphyrin-based molecules in a unique J-aggregated structure on the water surface. This is achieved by using charged surfactants as templating layers on the water surface through azimuthal alignment epitaxy, specifically *point-on-line* coincidence epitaxy. This unique J-aggregation structure on the water surface is substantially confirmed by different spectroscopy techniques (UV-vis, PL, and Raman spectroscopy), in-situ GIWAXS, and along with theoretical calculations.

3. Molecular-level mechanistic understanding of programmable assembly approach. We explored the stepwise mechanism to understand the unique selectivity and reactivity on the water surface by employing various in-situ techniques directly on the water surface, such as sum frequency generation (SFG) spectroscopy and grazing-incidence wide-angle X-ray scattering (GIWAXS). The dynamic interfacial surface charge density on the water surface is attributed to the formation of monolayer-to-multilayer, enabling the controllable growth of the site-selective product.

Reviewers' Comments:

Reviewer #1:

Remarks to the Author:

I'm pleased by the revisions the authors made, in relation to my comments as well as those of the other referees. This work is an important contribution.

Reviewer #3:

Remarks to the Author:

The manuscript has been improved.

I main concern is that this manuscript is a very technical paper, not really a scientific paper.

If the author claim "programmable assembly" is correct, the authors have to prove they are able to create different stuctures using this method. They only show one. The AFM image do not show ordered stuctures.

To my opinion, this paper is more suitable for a more specific journal.

Detailed response to the comments from the reviewers

Reviewer #1 (Remarks to the Author):

General comment: *I'm pleased by the revisions the authors made, in relation to my comments as well as those of the other referees. This work is an important contribution.*

Response: We greatly appreciate the Reviewer for the highly positive comments with recommendation for publication.

Reviewer #3 (Remarks to the Author):

General comment: *The manuscript has been improved. I main concern is that this manuscript is a very technical paper, not really a scientific paper. If the author claim "programmable assembly" is correct, the authors have to prove they are able to create different stuctures using this method. They only show one. The AFM image do not show ordered stuctures. To my opinion, this paper is more suitable for a more specific journal.*

Response: We highly appreciate the valuable comments from the reviewer. Our discovery highlights the fundamental understanding of on-water surface chemistry and also has significant implications for other fields. The key fundamental features of our study lie in:

- 1. Unique site-selectivity and reactivity in chemical reactions on the water surface.**
- 2. Surfactant-driven J-aggregated molecular confinement.**
- 3. Molecular-level mechanistic understanding of programmable assembly approach.**

In this work, we demonstrate the discovery of site-selective chemical reactions on the water surface and present the proof-of-concept by showing a wide range of reversible and irreversible chemical reactions. Overall, we illustrate three different chemical structures: SSC-1 (imide), SSC-2 (imine), and SSC-3 (1,3-diazole/imidazole). The programmable assembly approach is defined as a step-by-step assembly process that involves the pre-organization of the precursor compound, driving the organized structures of the final product.

We present time-dependent atomic force microscopy (AFM) measurements on the on-water surface film, subsequently transferred onto a SiO₂/Si substrate using the Langmuir-Schaefer method. The objective was to investigate the number of R1 layers adsorbed underneath the SOS surfactant monolayer and to observe the layer-by-layer growth of a multilayer, site-selective product resulting in a thickness of approximately 13 nm. The few-layer transferred sample

displayed a highly uniform appearance. However, as the thickness increased, the sample exhibited some roughness, as is often observed in wet transfer samples from the water surface.